# A comparative analysis of attitudes toward genome-edited food among Japanese public and scientific community

**Ryuma Shineha**[1]*, **Kohei F. Takeda**[1], **Yube Yamaguchi**[2], **Nozomu Koizumi**[2]

**1** Research Center on Ethical, Legal and Social Issues, Osaka University, Osaka, Japan, **2** Graduate School of Agriculture, Osaka Metropolitan University, Osaka, Japan

* shineha@elsi.osaka-u.ac.jp

**Data Availability Statement:** All relevant data are within the manuscript and its Supporting information files.

## Abstract

Genome editing technologies such as CRISPR/Cas9 have been developed in the last decade and have been applied to new food technologies. Genome-edited food (GEF) is a crucial issue with those new food technologies. Thus, each country has established GEF governance systems to maximize benefits and minimize risks. These emphasize the importance of communicating about GEF to the public. The key concerns are understanding various viewpoints and value perspectives (framings) in science and technology and encouraging and opening communication with the public. Thus, it is essential to understand differences between the public and experts' interests and discuss various framings and effective communication with regard to GEF. Accordingly, this study involved administering a questionnaire to analyze the public's attitudes in Japan and identify gaps between these and expert opinions on GEF. A total of 4000 responses from the public and 398 responses from GEF experts were collected. The study found that the Japanese public has a "wait-and-watch" attitude toward GEF, and the demand for basic information on it is quite high. Moreover, they are apprehensive about proper risk governance systems for GEF. This is despite experts' emphasis on the adequacy of the mechanism, necessity of technology, and trust in the scientific community. Understanding gaps between the public and experts' opinions on and interests in GEF provides essential insight for effective communication and acts as the basis for appropriate governance of emerging science and technology.

## Introduction

Genome editing technology such as CRISPR/Cas9, which received the Nobel Peace Prize in 2020, has been expected in new plant breeding techniques. Countries are attempting to streamline the framework of GEF regulation as new genome-edited foods (GEF) are being developed globally. There are several approaches to governance, focusing on "process" and "products." While the United States implements "product-based" regulations, the European Union implements overall strict regulations focusing on "process." This is similar in the case of genetically modified organisms (GMOs). However, after the United Kingdom (UK)

**Funding:** This study was supported by the MHLW Food Safety Promotion Research Program (PI: Nozomi Koizumi, 21KA1002), and the project titled "Implementation and systematization of RRI assessment model on emerging science and technology" (PI: Ryuma Shineha, JPMJRX20J2), funded by JST-RISTEX. Currently, KAKEN (PI: Ryuma Shineha, 23K17489) is supporting this analysis. The funders had no role in study design, data collection and analysis, decision to publish, or preparation of the manuscript.

**Competing interests:** The authors have declared that no competing interests exist.

underwent Brexit in 2021, it made its regulatory system "product-based," like the United States [1–9]. Tachikawa and Matsuo indicated four approaches to governance on GEF through international comparison [9]. First, GMO regulations are applied as they are to GEF. Thus, safety assessment and governmental approval are required (e.g., EU). The second is that simplified GMO regulations will be applied to GEF (e.g., UK, China). The third is slightly different from the former two approaches. Although genome-edited products are exempt from GMO regulations, governmental consultation or confirmation is required before placing them on the market (e.g., Japan, Argentina). Japan takes this third approach. In the Japanese case, notification of information concerning GEF is also conducted. Fourth, genome-edited products are exempt from GMO regulations, and prior confirmation is not required by the government (e.g., US) [9].

In Japan, the GEF regulatory framework has been discussed actively, especially after 2018. Several political documents on science and innovation policies in biotechnologies have been published [9–12]. They attach high expectations to genome editing and encourage research and development through policy-support and funding. They are often performed and driven by notions and discourses of an "all Japan effort" [13].

Concerning the food application of GEF, it is regulated by the Ministry of Health, Labour, and Welfare (MHLW). The food application of GEF is governed by the Food Sanitary Law. Their treatment of GEF also depends on the type of GEF. Although there is little difference in field cultivation, the food application of GEF is divided into two types. If the GEF is regarded as non-Genetically Modified (GM) food, the GEF provider will be expected to notice the safety of their products and cooperate to share information on environmental effects. However, it is not legally mandatory. If the GEF is categorized as GM food, it will be evaluated equally to other GM foods. Generally, product-based approaches are adopted. In both cases, providers are required to conduct consultation with the MHLW [9,12–15].

Within the Japanese regulatory system on research and development, genome editing is differently controlled by three types of editing: SDN-1, SDN-2, and SDN-3. SDN-1 is the technique of introducing a break at a target site in the genome. SDN-2 is the addition of a few base pairs of nucleotides to the target site. SDN-3 is introducing a gene at the target site. These types of gene editing are differently regulated according to the viewpoints of environmental assessment and food sanitation [9,12].

For field cultivation of genome-edited crops, they are regulated mainly by the Ministry of Environment (MoE) and Ministry of Agriculture, Forestry, and Fishery (MAFF), and their discussions are related to issues of environmental assessment and biodiversity. In this field cultivation, categories of S-2 and SDN-3 of genome-edited crops (or animals) are regulated under the Cartagena Domestic Law enacted in conformance with the Cartagena Protocol [9,12].

Although the governance of GEF is developed based on the global progress of research and development described above, commonly, the GEF governance model emphasizes the significance of communicating GEF to the public, considering lessons from previous GMO cases. Currently, discussions on governing emerging science and technology take the perspective of responsible research and innovation (RRI) [16,17]. The framework examines ethical, legal, and social issues (ELSI), upstream public engagement, and the inclusion of various stakeholders in modern innovative processes. RRI has been expressed as "responsible innovation means taking care of the future through collective stewardship of science and innovation in the present" [17]. The key issues herein are adequate understanding of various framings of science and technology and encouragement and opening up of communication with the public. Therefore, it is essential to understand differences between the public and expert's interests when discussing various framings and effective communication on GEF.

The earlier context of GMO involved trials on public engagement [18–26]. A famous trial in the UK, titled "GM Nation? The Public Debate," revealed various public interests and framings, such as concerns regarding risks (other than scientific ones), risk-benefits, critical perspectives on globalization, alibi-building, distrust in governments and multinationals, fairness in trade, food distribution, and will to further discussions [20–26]. Similarly, trials on public engagements with GEF have been conducted. In the Japanese case, these revealed public interests in safety, environmental effects, food distribution, transparency of information, bioethics, right to choose, and the future of local agriculture [27,28].

In several previous studies, questionnaires on public attitudes toward biotechnologies, including GMOs, were administered. Even after considering the effect of scientific literacy, they found that gaps between self-confidence in knowledge and literacy significantly influence public acceptance of GMOs, more than literacy [29]. Generally, the recognition of various social, psychological, and cultural factors, such as "trust" in science and "morality," influence social acceptance of GMOs more significantly [30–37]. Regarding new breeding technologies, their analysis of data from European and North American expert panels indicated the necessity of considering regulations, political attitudes and structures, and social and environmental concerns, for accelerating new breeding technologies [38,39].

Similarly, in Japan, several questionnaire-based studies have investigated public attitudes, and acceptance of GMO and GEF. In the case of GMO, previous studies found that, generally, the Japanese public has negative impressions and responses [40]. Recently, a research group compared public attitudes (N = 3000) and genome editing experts' opinions (N = 197), toward the applications of three agricultural biotechnologies: conventional breeding, GMO, and GEF. Public has stronger interests in risks, safety, and mandatory labeling of GEF, as compared to experts [41]. Interestingly, the public had more positive attitudes toward GEF than GMO. The group also examined public perceptions of the benefits and risks of these three agricultural biotechnologies by comparing the before and after of sharing information on scientific mechanisms. This revealed that sharing information increases and decreases the perception of benefits and risks, respectively. However, this trend is most prominent in the case of conventional breeding. The research group also found that, while scientific literacy increases perceptions of benefits, it does not decrease perceptions of risks. Thus, they emphasized the complexity and uncertainty of public attitudes toward emerging breeding technologies [42,43].

A recent literature, after the onset GEF commercialization, showed consumer acceptance of GEF—with 550 respondents [44]. It was found that an interaction existed between awareness of genome-editing technology, information credibility, perceived usefulness, and willingness to buy GEF [44].

However, these studies did not arrive at the difference in interests and framings of communication between the public and experts. Previous studies on the Japanese public's attitudes toward Stem Cell Research (SCR) and Regenerative Medicine (RM) found differences in communicative interests and social acceptance between the public and experts. For instance, while the public was primarily interested in the consequences of successful RM (such as cost), counter-measures for risks and accidents, and responsibility and liability, the RM experts focused on scientific content and validation, considering the lack of clinical RM trials [45]. These tendencies are replicated in South Korea [46].

Considering these existing studies, this study examined differences in interests between the public and experts. The key questions were:

- What is the overview of public attitudes toward GEF in Japan?

- Are there different interests in communication between the public and experts regarding what is sought to be known and what is informed?

- Are there any differences in factors for social acceptance of GEF between the public and experts?

## Methods

From February 21 to April 2, 2020, survey questionnaires were circulated among the Japanese public and experts in the GEF community through the internet. They were informed that they would participate in the survey using the internet and that all data would be de-identified and only reported in the aggregate. All participants acknowledged an informed consent statement in order to participate in the study through the Rakuten Insight Co. A total of 4000 responses were obtained from participants from the general public, recruited through Rakuten Insight ©. There was an equal number of participants from a particular age group (from 20- to 70-year-olds) and gender (male or female). Simultaneously, 398 valid responses were acquired from experts, upon circulating the questionnaire via mailing lists of the Japanese Society of Genome Editing, and other related groups. The demographic distribution of respondents is presented in Table 1.

We formulated multiple-choice questions to conveniently compare answers from the public and experts. Before the public participants answered the questions, they were given brief explanatory information on GEF. The most important questions were: "Which topics do you want to know about/ be informed of, regarding GEF?" and "Which factors are important for

**Table 1. Demographic distribution of respondents.**

|  | The Public | Expert |
|---|---|---|
| **Gender** |  |  |
| Male | 50.0% | 78.80% |
| Female | 50.0% | 21.20% |
| Other/NA | – |  |
| **Age** |  |  |
| Average age | 49.3 | 43.1 |
| **Education** |  |  |
| High school or less, other | 31.8% | 1.3% |
| Vocational college | 11.8% | 0.0% |
| Junior college | 10.0% | 0.0% |
| College | 41.9% | 5.0% |
| Graduate school | 4.6% | 93.7% |
| **Educated field** |  |  |
| Science | 33.9% | 98.5% |
| Humanities and Social Science | 60.3% | 1.5% |
| Other | 5.9% | 0.0% |
| **Annual Income** |  |  |
| Under 4 million JPY | 36.3% | – |
| 4 million JPY to 6 million JPY | 24.0% | – |
| 6 million to 8 million JPY | 17.0% | – |
| Over 8 million JPY | 22.8% | – |
| N | 4000 | 398 |

**Table 2. Basic structure of the questionnaire.**

| Theme | Focused themes | References on previous studies |
|---|---|---|
| **Recognition of Genome Editing Foods** | Recognition of keywords in genome editing foods | [37,41–43,45,46,49] |
| | Opinion on the promotion of genome editing foods | |
| | Opinions on regulations for genome editing foods | |
| | Opinions on the labeling of genome editing foods | |
| | Trust in experts' stories on safety | |
| **Interest topics on Genome Editing Foods** | Interested topics | [45–48] |
| | Important factors for social acceptance | |
| **Demography** | Age, Gender, Education, Income, Expertise, etc. | |

the social acceptance of GEF?" These questions were formulated on the basis of previous studies on public attitudes toward SCR and RM [45,46], and rooted in a large-scale questionnaire on nuclear energy [47,48]. Additionally, we sought opinions on keyword recognition, labeling issues, expected regulation, etc. [37,41–49].

Table 2 presents a rough sample of the questionnaire (for details, see Sup. 1). A cross-table analysis was conducted for comparing differences in the public's and experts' answers.

In our statistical analysis, we conducted a chi-square test to determine independence. In addition, we calculated Pearson's correlation (see S1 Table). For our analysis, we used SPSS ver. 26.

## Results

### Recognition of genome-edited food (GEF)

Fig 1 presents public responses to questions on recognizing GEF. Fig 1(a) presents that more than half of the respondents (55.7%) answered that "I have heard of it, but am not familiar with its contents," while 28.0% answered that "I have never heard of it." This indicates that GEF is yet to be common in Japanese society.

In response to the social acceptance of GEF, 43.6% of the respondents were undecided, 29.9% found it acceptable, while 26.5% found it unacceptable [Fig 1(b)]. Even with regard to consuming GEF, 60.8% of the respondents were undecided [Fig 1(c)]. Responses to questions on trusting experts also varied [37]. In response to the question, "Can you trust experts on the safety of GEF?," 52.2% of the respondents were undecided, 17.3% "agreed," and 30.5% "disagreed" [Fig 1(d)]. Generally, the Japanese public was hesitant to show concrete attitudes toward GEF.

There are strong correlations between the answer pattern of the three questions on attitudes toward GEF (See S1 Table). Generally, the stronger respondents think GEF will be accepted in society, the stronger they agree with eating GEF. Simultaneously, they also show their trust in expert's discourses on the safety of GEF.

Fig 2 presents a comparison of public and expert opinions on regulating and developing GEF. With regard to GEF labeling, 65.2% of the public answered that "Genome-edited foods must be mandatorily labeled." More than half of the experts (51.3%) also stated the same. However, 18.1% of the experts did respond that "Labeling genome-edited food is not necessary." This differs from the public response to this (1.4%) [Fig 2(a)]. Fig 2(b) presents that

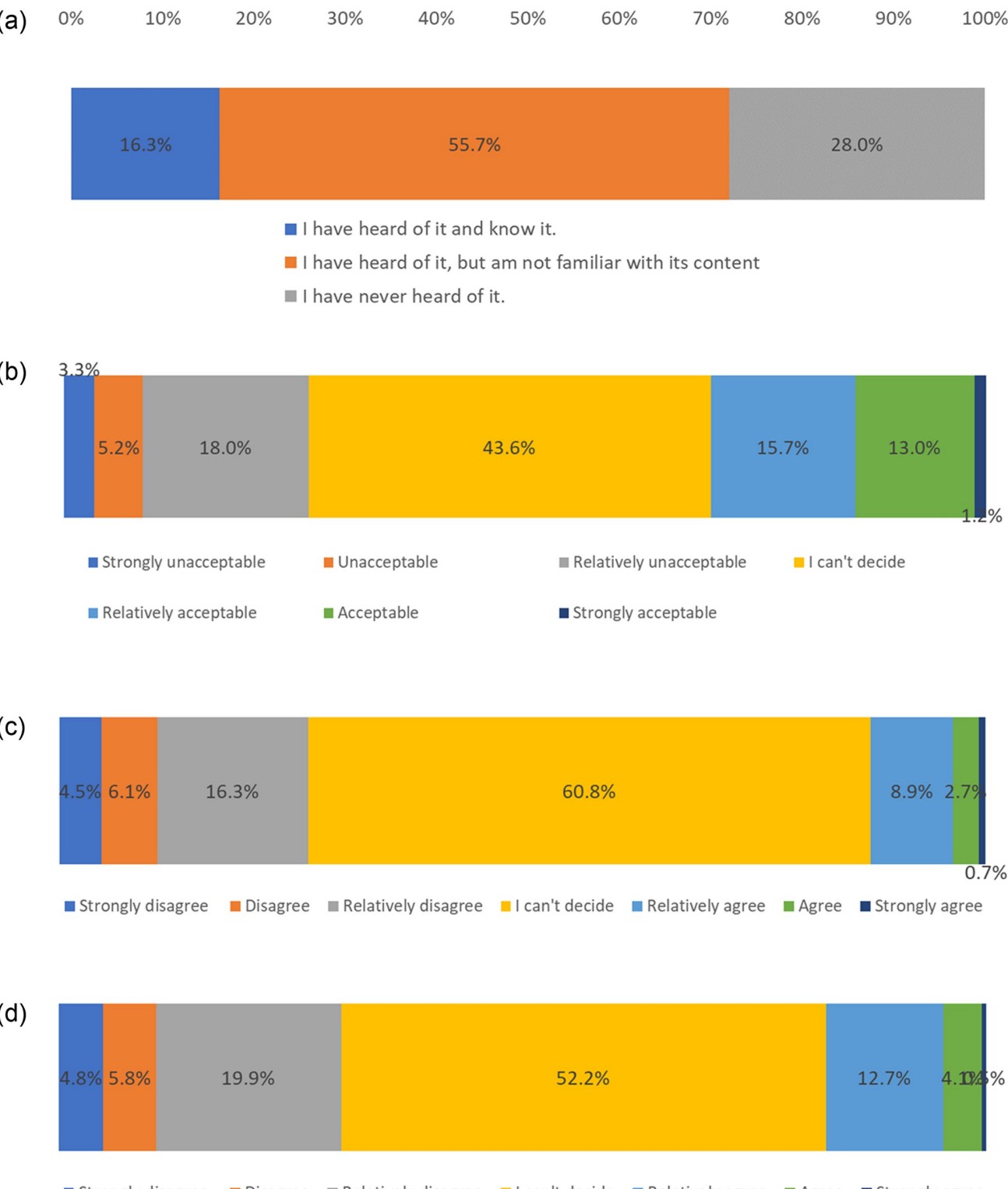

**Fig 1. Public recognition of genome-edited food (GEF).** (a). Recognition of keywords related to genome-edited food (GEF) (Have you heard of keywords related to GEF?). (b). Opinions on the social acceptance of genome-edited food (GEF) (Do you think that GEF will be accepted in society?). (c). Degree of agreement with eating genome-edited food (GEF) (Do you agree with eating GEF?). (d). Trust-worthiness of expert's discourses on safety of genome-edited food (GEF) (Can you trust experts' discourses on the safety of GEF?).

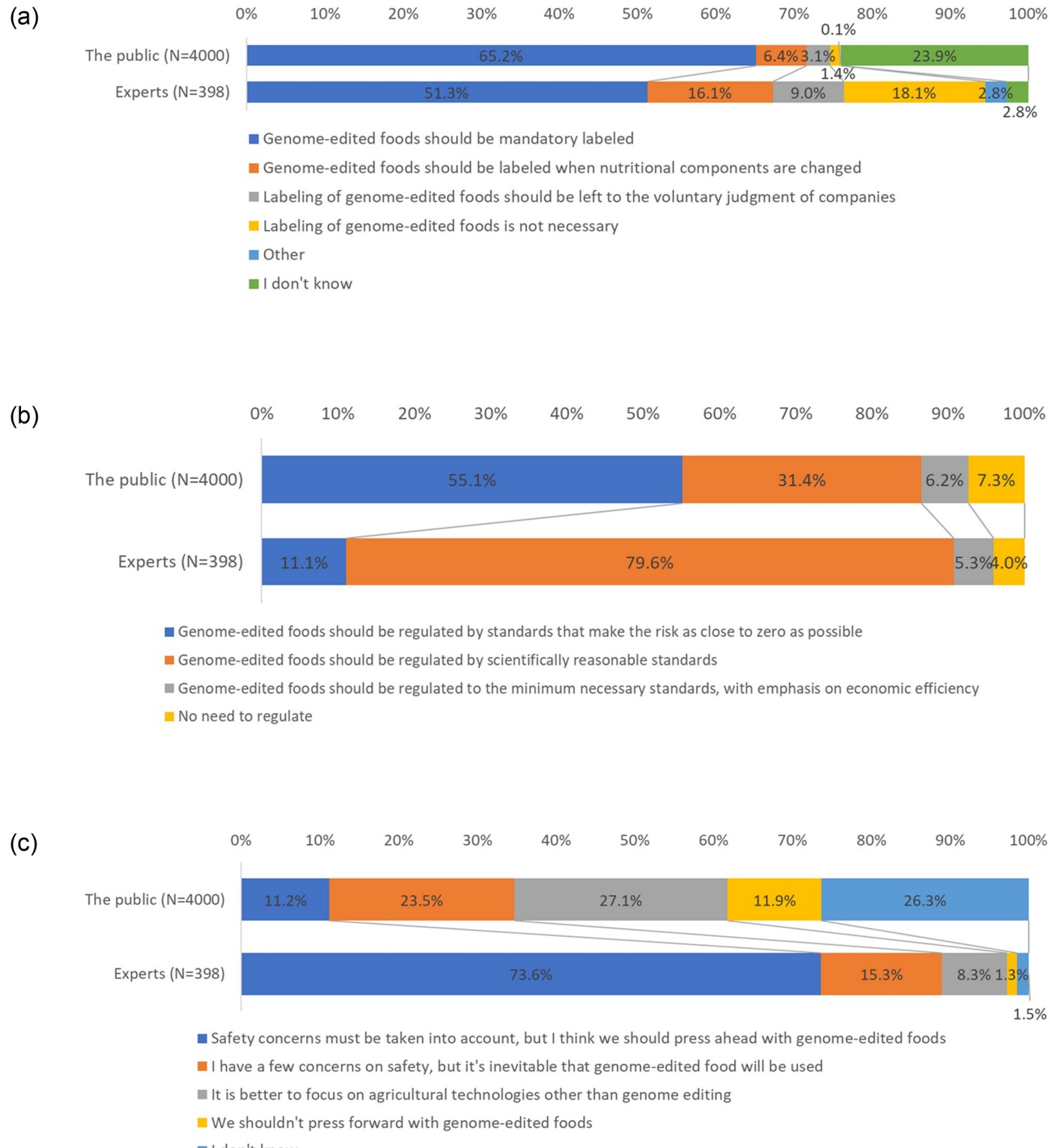

**Fig 2. Comparison of public and expert opinions on genome-edited food (GEF).** (a). Opinions on labeling genome-edited food (GEF) (Please select one statement from the following that best describes your thoughts about labeling GEF). (b). Opinions on regulating genome-edited food (GEF) (Please select one statement from the following that best describes your thoughts about regulating GEF). (c). Opinions on the progress of genome-edited food (GEF) (Please select one statement from the following that best describes your overall thoughts about GEF).

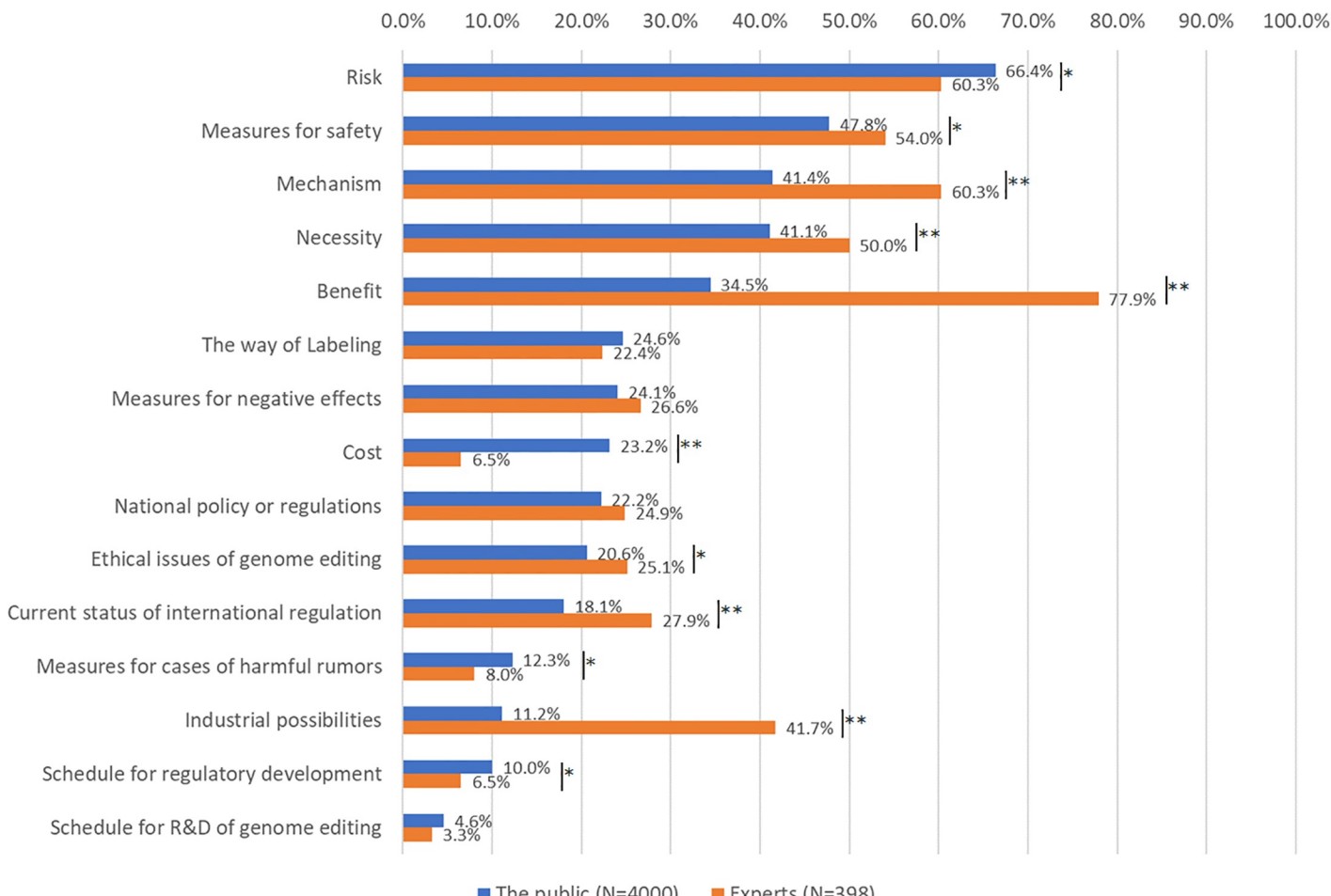

**Fig 3. Topics of interest with regard to genome-edited food (GEF) (The public: What do you want to know? Experts: What do you want to inform? Please choose three interesting topics).** A chi-square test was conducted with *p < 0.05, **p < 0.01.

public and expert opinions on the GEF regulatory framework also differ. While 55.1% of the public answered that "Genome-edited food should be regulated by standards that reduce risk to as close to zero as possible," most experts answered that "Genome-edited food should be regulated by scientifically reasonable standards" (79.6%). Moreover, opinions on the development of GEF differed. Most experts (73.6%) answered that "Although safety concerns must be taken into account, we should press ahead with genome-edited food." However, the public stated that "I have a few concerns with safety, but it is inevitable for genome-edited food to be consumed" (23.5%), and "It is better to focus on agricultural technologies other than genome editing" (27.1%).

## Differences in public and expert interests in genome-edited food (GEF)

The topics of interest to both the public and experts were comparatively analyzed (Fig 3). The public was questioned about the topics they required knowledge on, while experts were questioned on those they wanted to educate the public on. The analysis involved a chi-square test to determine independence. Interestingly, although the order differed, the top five topics of interest were common to the public and experts. These included: "risk" (the public: 66.4%,

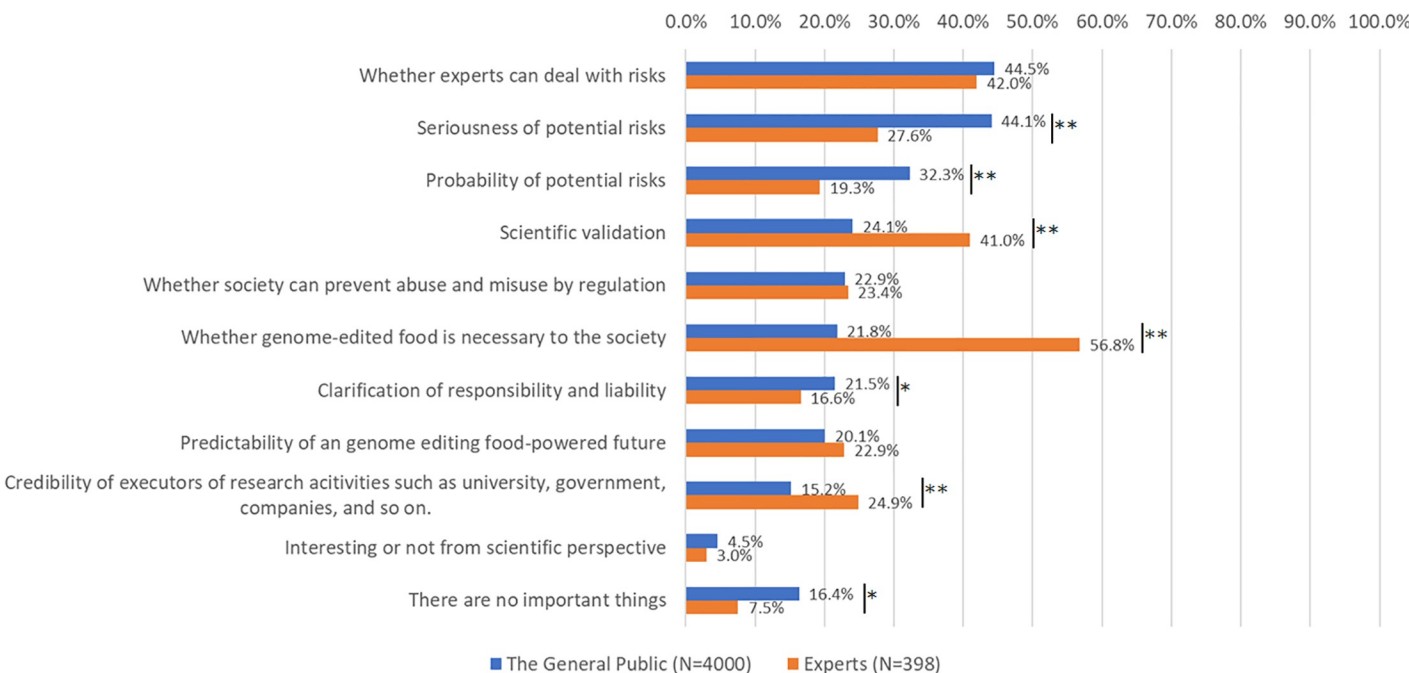

**Fig 4. Comparison of factors important to the public and experts, for the social acceptance of genome-edited food (GEF) (Which factors are important for the social acceptance of GEF? Please choose three factors).** A chi-square test was conducted with *p < 0.05, **p < 0.01.

experts: 60.3%), "measures for safety" (the public: 47.8, experts: 54.0%), "mechanism" (the public: 41.4%, experts: 60.3%), "necessity" (the public: 41.1%, experts: 50.0%), and "benefit" (the public 34.5%, experts: 77.9%).

It is note-worthy that the degree of interest in these topics differed; the experts emphasized more on "benefit," "mechanism," "necessity," "industrial possibilities" (the public: 11.2%, experts: 41.7%), and "current status of international regulation" (the public: 18.1%, experts: 27.9%) (*p* < 0.01). The values for "measures for safety" and "ethical issues in genome editing" (the public: 20.6%, experts: 25.1%) also demonstrated similar trends (*p* < 0.05). However, the public emphasized on "cost" more than experts (the public: 23.2%, experts: 6.5%) (*p* < 0.01). They were also more interested in "risk," "measures for cases of harmful rumor" (the public: 12.3%, experts: 8.0%), and "schedule for regulatory development" (the public: 10.0%, experts: 6.5%) (*p* < 0.05).

Fig 4 presents important factors for the social acceptance of GEF. These too differ for the public and experts. Largely, the public regarded risk governance factors as important for the social acceptance of GEF. These included: "can experts deal with the risks" (the public: 44.5%, experts: 42.0%), "seriousness of potential risks" (the public: 44.1%, experts: 27.6%, *p* < 0.01), "probability of potential risks" (the public: 32.3%, experts: 19.3%, *p* < 0.01), and "clarification of responsibility and liability" (the public: 21.5%, experts: 16.6%, *p* < 0.05).

While the public emphasized on factors related to GEF risks, experts focused on factors concerning the social necessity of, and trust in, scientific communities, for the social acceptance of GEF. These included: "Is GEF necessary for society" (the public: 21.8%, experts: 56.8%, *p* < 0.01), "scientific validation" (the public: 24.1%, experts: 41.0%, *p* < 0.01), and "credibility of research executors such as universities, governments, companies, etc." (the public: 15.2%, experts: 24.9%, *p* < 0.01).

**Table 3. Cross tabulation of public responses between questions on the trust-worthiness of experts on GEF and important factors for acceptance.**

| Important factors for acceptance | Can you trust the story of experts on the safety of genome edited foods? | | | | | | |
|---|---|---|---|---|---|---|---|
| | Strongly disagree | Disagree | Relatively disagree | I can't decide | Relatively agree | Agree | Strongly agree |
| Whether experts can deal with risks | 30.2% | 47.4% | 52.6% | 42.1% | 48.5% | 38.4% | 28.6% |
| Seriousness of potential risks | 42.7% | 57.4% | 56.3% | 39.2% | 43.4% | 35.4% | 14.3% |
| Probability of potential risks | 29.7% | 36.1% | 37.6% | 29.4% | 35.1% | 33.5% | 19.0% |
| Scientific validation | 6.8% | 12.2% | 21.2% | 22.1% | 37.5% | 55.5% | 52.4% |
| Whether society can prevent abuse and misuse by regulation | 13.5% | 23.5% | 24.0% | 19.8% | 32.0% | 37.2% | 38.1% |
| Whether genome editing food is necessary to the society | 8.9% | 20.9% | 20.1% | 21.6% | 27.2% | 30.5% | 38.1% |
| Clarification of responsibility and liability | 21.9% | 24.3% | 27.2% | 19.7% | 20.3% | 17.7% | 9.5% |
| Predictability of an genome editing food-powered future | 20.8% | 26.5% | 23.5% | 17.7% | 21.1% | 20.1% | 19.0% |
| Credibility of executors of research activities such as university, government, companies, and so on. | 12.0% | 17.4% | 16.8% | 12.7% | 22.1% | 17.7% | 19.0% |
| Interesting or not from scientific perspective | 2.6% | 0.4% | 2.3% | 4.0% | 8.7% | 10.4% | 61.9% |
| There is no important thing | 37.0% | 11.3% | 6.1% | 23.9% | 1.4% | 1.2% | 0.0% |

We also conducted cross tabulation of public responses between questions on the trust-worthiness of experts on GEF (see Fig 1(d)), important factors for acceptance, and interested topics (see Tables 3 and 4). Generally, we found that people who trust expert's discourses on safety regard "scientific validation," "necessity," and "the prevention of abuse and misuse by regulation" as key factors for social acceptance of GEF. They also would like to know the "mechanism," "benefit," and "cost" compared to another group. However, people who do not trust expert's discourses on safety regard "risk" and "clarification of responsibility and liability" as key factors for social acceptance of GEF. They would like to know "measures for negative effects," "national policy or regulation," and "ethical issues."

## Discussion

This study found that the public reserved its judgements on GEF (Fig 1). This implies attitudes of "wait-and-watch," and not necessarily negative toward it (Fig 1). This is relatively different than the case of GMO, which received clear negative responses [12]. Such relatively neutral responses to GEF were also seen in previous Japan-based studies [42,43]. Our results indicate similar trends. This study found that the public expects strict regulation of GEF as compared to experts (Fig 2). Such gaps between public and expert opinion on GEF coincide with the findings of previous studies [41].

Interestingly, this study's results indicate that a section of the public believes that "It is better to focus on agricultural technologies other than genome editing" [Fig 2(c)]. Previous studies had found that, upon being informed about technological mechanisms, conventional breeding was perceived most positively when GMO and GEF were compared [42,43]. If we interpret our results in light of these previous findings, GEF emerges as just one of the agricultural technology options available to consumers. It appears that visions of a combination between GEF and traditional agriculture may bring new public attitudes. At least the fact that GEF is not exclusively related to conventional breeding must be explained. Moreover, a vision of agriculture wherein GEF and conventional breeding complement each other, considering public interests in local agricultural contexts and cultures, must be presented [27,28].

For more effective communication between the public and experts on GEF, we must understand the points of interest to both parties. According to our findings, both parties shared a

**Table 4. Cross tabulation of public responses between questions on the trust-worthiness of experts on GEF and interested topics.**

| What do you want to know | Can you trust the story of experts on the safety of genome edited foods? | | | | | | |
|---|---|---|---|---|---|---|---|
| | Strongly disagree | Disagree | Relatively disagree | I can't decide | Relatively agree | Agree | Strongly agree |
| Risk | 45.8% | 65.2% | 73.4% | 61.4% | 80.7% | 76.8% | 61.9% |
| Measures for safety | 35.9% | 46.1% | 52.9% | 44.7% | 57.4% | 50.6% | 23.8% |
| Mechanism | 21.4% | 32.6% | 42.3% | 41.1% | 48.7% | 50.0% | 66.7% |
| Necessity | 28.1% | 37.4% | 45.5% | 40.5% | 42.6% | 40.9% | 47.6% |
| Benefit | 10.9% | 20.4% | 33.0% | 33.3% | 49.9% | 52.4% | 66.7% |
| The way of Labeling | 21.9% | 33.0% | 30.2% | 20.5% | 28.2% | 29.9% | 23.8% |
| Measures for negative effects | 19.8% | 32.2% | 30.4% | 20.7% | 25.0% | 26.8% | 28.6% |
| Cost | 7.8% | 11.7% | 16.8% | 24.3% | 33.9% | 36.6% | 52.4% |
| National policy or regulations | 17.7% | 23.9% | 26.7% | 20.2% | 25.0% | 22.0% | 9.5% |
| Ethical issues of genome editing | 26.0% | 30.9% | 25.5% | 16.6% | 23.3% | 21.3% | 4.8% |
| Current status of international regulation | 17.7% | 17.8% | 24.2% | 15.2% | 18.9% | 22.0% | 19.0% |
| Measures for cases of harmful rumors | 13.0% | 13.0% | 14.7% | 10.6% | 12.8% | 16.5% | 33.3% |
| Industrial possibilities | 5.2% | 8.3% | 9.4% | 10.0% | 17.9% | 20.7% | 38.1% |
| Schedule for regulatory development | 10.9% | 12.6% | 11.4% | 8.5% | 12.8% | 9.8% | 9.5% |
| Schedule for R&D of genome editing | 2.6% | 5.2% | 4.3% | 3.8% | 5.9% | 11.6% | 14.3% |

combination of their top five concerns (Fig 3). It is important to provide basic information on GEF to the public continuously, considering the potential effects of information-sharing on the mechanisms and benefits of GEF, as suggested by previous studies [44]. With regard to factors behind the social acceptance of GEF, although regulation was similar between the public and expert, experts emphasized social necessity and trust in scientific communities, while the public focused on risk-related factors (Fig 4).

These results differ slightly from those of previous studies on SCR and RM [45]. Both cases displayed a common trend: high interest in "risks" related to technology and emphasis on "scientific validation" by experts. In the cases of SCR and RM, the public was more interested in how the technology would be handled post-implementation, such as in responses to medical accidents, responsibility and liability, and prevention of abuse and misuse. In this study, although the public was found to be more interested in such themes than the experts, these were less prioritized than in the SCR and RM cases. It can be interpreted that the public was more focused on the "risk governance" of GEF.

Effective and real-time communication, based on a better understanding of gaps between public and expert interests, would increase "informed-ness," which is a key factor in attitudes toward new biotechnologies [30]. It shall serve as the basis of trust [37]. These are essential for improving RRI governance systems for emerging science and technology. The results of this study shall contribute to related discussions. Simultaneously, to enable a continuous dialogue among all stakeholders, we should not only consider differences in interests found by this and previous studies but also hurdles in and measures for encouraging communication between the public and experts. Time, opportunities, and evaluation systems for such communicative activities are important factors in encouraging experts to communicate [50].

Simultaneously, we need to consider the different attitudes between the two types of groups, with or without trust in experts' discourses on safety. Tables 3 and 4 show that the two groups have different interests and concerns regarding GEF. People who trust expert's discourses on safety generally have interests in scientific contents and "the way of prevention of abuse and misuse by regulation." On the other hand, another group have interests in issues related risk

governance such as "whether experts can deal with risks," "clarification of responsibility and liability," "national policy or regulation," "ethical issues," and so on. We should not see the public as a monolith. We need to consider what kinds of information and factors are of interest and expected in each group.

Our study poses the following question for future research: "What are the factors and roots of different trends in public attitudes toward GEF, as compared to GMO, SCR, and RM?" Potential factors include the spread of keywords, differences in technologies, public imaginaries, media discourses, etc. We need to examine how these factors influence the perception of each type of technology. With the greater anticipation of genome editing, communication that accounts for diverse public opinions, affected by political, cultural, social, ethical, and moral factors, has become a prominent issue.

This study faced certain limitations. First, responses were collected via a web-based questionnaire, and public participants were recruited through Rakuten Insight ©. Overall, their income was higher than the Japanese average (approximately 4.43 million JPY, also see Table 1). Hence, certain sampling biases could not be avoided. Second, although more experts participated in this study (N = 398) than in previous ones [41–43], a sampling bias may have been present due to a relative lack of voluntary participants. Thus, we would not stress upon the statistical representativeness of our data, particularly experts' data. Another limitation is the timing of our survey. After our survey, the commercialization of GEF, such as genome-edited tomato and fish, started in Japan. Thus, we need continuous monitoring of public attitudes toward GEF to discuss the latest trend. However, we do believe that our data shall act as a reference for understanding public attitudes toward GEF and different framings of communication between the public and experts.

## Conclusions

This study found that the Japanese public has a "wait-and-watch" attitude toward GEF, and the demand for basic information on it is quite high. They also anticipate proper risk governance systems for GEF. However, their attitude toward the social acceptance of GEF differs from that of experts. While experts emphasize the mechanism, necessity of technology, and trust in the scientific community, the public is more interested in risk governance. Understanding gaps between public and expert opinions and interests regarding GEF is essential for effective communication and shall further serve as the basis for appropriate governance of emerging science and technology. Our results shall contribute to this discussion.

## Supporting information

**S1 File.**
(DOCX)

**S1 Dataset.**
(SAV)

**S1 Table. Peason's correlation between the attitudes toward GEF.**
(DOCX)

## Acknowledgments

We would like to thank Editage (www.editage.jp) for English language editing.

## Author Contributions

**Conceptualization:** Ryuma Shineha, Yube Yamaguchi, Nozomu Koizumi.

**Data curation:** Ryuma Shineha, Kohei F. Takeda, Yube Yamaguchi.

**Formal analysis:** Ryuma Shineha, Kohei F. Takeda.

**Funding acquisition:** Ryuma Shineha, Nozomu Koizumi.

**Investigation:** Ryuma Shineha, Nozomu Koizumi.

**Methodology:** Ryuma Shineha.

**Project administration:** Ryuma Shineha.

**Supervision:** Ryuma Shineha, Nozomu Koizumi.

**Writing – original draft:** Ryuma Shineha.

**Writing – review & editing:** Kohei F. Takeda, Yube Yamaguchi, Nozomu Koizumi.

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
