## [Decision Letter · Decision Letter 0]

8 Nov 2023

PONE-D-23-28736A comparative analysis of attitudes towards genome-edited food among Japanese public and scientific communityPLOS ONE

Dear Dr. Shineha,

Thank you for submitting your manuscript to PLOS ONE. After careful consideration, we feel that it has merit but does not fully meet PLOS ONE’s publication criteria as it currently stands. Therefore, we invite you to submit a revised version of the manuscript that addresses the points raised during the review process.

We look forward to receiving your revised manuscript.

Kind regards,

Basavantraya N. Devanna, PhD

Academic Editor

PLOS ONE

Journal Requirements:

2. You indicated that ethical approval was not necessary for your study. We understand that the framework for ethical oversight requirements for studies of this type may differ depending on the setting and we would appreciate some further clarification regarding your research. Could you please provide further details on why your study is exempt from the need for approval and confirmation from your institutional review board or research ethics committee (e.g., in the form of a letter or email correspondence) that ethics review was not necessary for this study? Please include a copy of the correspondence as an ""Other"" file.

A clean copy of the edited manuscript (uploaded as the new *manuscript* file)”"

"This study was supported by the MHLW Food Safety Promotion Research Program (PI: Nozomi Koizumi, 21KA1002), and the project titled “Implementation and systematization of RRI assessment model on emerging science and technology” (PI:Ryuma Shineha, JPMJRX20J2), funded by JST-RISTEX. Currently, KAKEN (PI: Ryuma Shineha, 23K17489) is supporting this analysis. We would like to thank Editage (www.editage.jp) for English language editing."

"This study was supported by the MHLW Food Safety Promotion Research Program (PI: Nozomi Koizumi, 21KA1002), and the project titled “Implementation and systematization of RRI assessment model on emerging science and technology” (PI: Ryuma Shineha, JPMJRX20J2), funded by JST-RISTEX. Currently, KAKEN (PI: Ryuma Shineha, 23K17489) is supporting this analysis. We would like to thank Editage (www.editage.jp) for English language editing. The funders had no role in study design, data collection and analysis, decision to publish, or preparation of the manuscript."

7. Your ethics statement should only appear in the Methods section of your manuscript. If your ethics statement is written in any section besides the Methods, please move it to the Methods section and delete it from any other section. Please ensure that your ethics statement is included in your manuscript, as the ethics statement entered into the online submission form will not be published alongside your manuscript.

**Additional Editor Comments:**

I thank the authors for this interesting article. However, before proceeding further, please address the queries raised by the reviewers.

Reviewers' comments:

Reviewer's Responses to Questions

**Comments to the Author**

1. Is the manuscript technically sound, and do the data support the conclusions?

Reviewer #1: Partly

Reviewer #2: Yes

2. Has the statistical analysis been performed appropriately and rigorously? 

Reviewer #1: Yes

Reviewer #2: Yes

3. Have the authors made all data underlying the findings in their manuscript fully available?

Reviewer #1: No

Reviewer #2: No

4. Is the manuscript presented in an intelligible fashion and written in standard English?

Reviewer #1: No

Reviewer #2: Yes

5. Review Comments to the Author

Reviewer #1: This manuscript describes a survey conducted on 4,000 Japanese people not involved in the genome editing field of work, representing the public group, and 398 Japanese people working in the genome editing field, representing the expert group. The objective of the survey was to obtain an overall view of the perception and acceptance of genome-edited food (GEF) in the Japanese market by both groups, and to compare their opinions, concerns and interests with regards to GEF. The authors found that while both the public and expert groups share the same concerns and interests in GEF, the general public is more indifferent in their perception and acceptance of GEF.

The authors successfully recruited independent groups of respondents from reliable sources. The public respondents represented a good spread of the society in terms of age, gender, education, and occupation/field of work, whereas the expert group was from an established society. The survey was informative, however, there are some missed opportunities in data analyses as highlighted below.

Comments:

In general, the intricacy of the survey is not clearly transmitted. Perhaps the data presentation is ambiguous but the depth of questions asked needs to be improved. For instance, is the question from Fig 1a “Recognition of keywords related to GEF (have you heard of keywords related to GEF?)” the only question asked to determine the public’s recognition of GEF or were there follow-up questions to assess the respondents’ responses? In addition, it is not clear if the respondents were given a list of keywords which they were asked to recognize, or were the respondents asked if they had heard of any keywords related to GEF? Similarly, in Fig 2, were the respondents given an example of a label or information that should appear on a label? A full version of the (translated) survey should be made available, i.e., in supplemental data for readers to better grasp the questions asked, hence the extent of the study conducted.

It would be informative to also have a breakdown of the percentage of respondents leading from one question to another to improve the depth of analysis. For instance, what percentage of respondents who “cannot decide” the social acceptance of GEF (Fig 1b) would or would not consume GEF (Fig 1c), and of which, how many were related to trust issues (Fig 1c)? Additionally, how does the degree of trustworthiness (Fig 1c) relate to the information the respondents are interested in (Fig 3)? What is the percentage of well-informed public are acceptive of GEF? Drawing some careful correlation analysis might provide more information and improve discussion.

Since the Japanese Society of Genome Editing is open to both academic and industry experts, authors should indicate if the experts are from the public or private sector. Reanalysis of the data taking the private or public sector into consideration would provide another perspective. The reason for this is to take into consideration the possibility of bias, i.e., that the private sector could be profit-driven.

In agreement with the authors’ awareness of the limitations of this survey, perhaps difficult but a follow-up on the public’s acceptance of GEF after the release of GEF products into the Japanese market would be highly interesting. While this request might be out of scope, discussion towards this direction would be appreciated. A helpful literature: https://doi.org/10.3390/su15129662

Minor:

Line 1: Nobel prize

Introduction could be improved, e.g. line 67-71: More elaboration on what are the 3 categories of GEF, and what are the 2 categories of regulation- how are they differently regulated?

Table 1: Under Education “High/Graduate School” should be “High/Graduate school”. Under Annual Income, “4 millionn” should be “4 million”

In general, language support would facilitate a smoother reading process.

Reviewer #2: Thanks for choosing me to review a research article entitled “A comparative analysis of attitudes towards genome-edited food among Japanese public and scientific community”, by Ryuma Shineha et. al. I read an article thoroughly and find it very interesting. It is related to public and experts’ perceptions about Genome edited food (GEF) in Japan. The sample size is large to conclude.

1. In abstract, a number of general public is given 3000, whereas this number is 4000 in the methods. Please correct it.

2. Correct Important word in Table 2.

6. PLOS authors have the option to publish the peer review history of their article (what does this mean?). If published, this will include your full peer review and any attached files.

Reviewer #1: No

Reviewer #2: No

---

## [Author Response · Author response to Decision Letter 0]

13 Jan 2024

Response to reviewer’s comment

We would like to appreciate to editors and reviewers for their kind cooperation and valuable opinions to our manuscript. We would like to show summary of our revision for reviewer’s comment here.

Reviewer 1:

1. In general, the intricacy of the survey is not clearly transmitted. Perhaps the data presentation is ambiguous but the depth of questions asked needs to be improved. For instance, is the question from Fig 1a “Recognition of keywords related to GEF (have you heard of keywords related to GEF?)” the only question asked to determine the public’s recognition of GEF or were there follow-up questions to assess the respondents’ responses? In addition, it is not clear if the respondents were given a list of keywords which they were asked to recognize, or were the respondents asked if they had heard of any keywords related to GEF? Similarly, in Fig 2, were the respondents given an example of a label or information that should appear on a label? A full version of the (translated) survey should be made available, i.e., in supplemental data for readers to better grasp the questions asked, hence the extent of the study conducted.

Answer

In order to make clear the contents and structure of questionnaire, we added English translation of key questionnaire (tentative version) as supplemental file.

2. It would be informative to also have a breakdown of the percentage of respondents leading from one question to another to improve the depth of analysis. For instance, what percentage of respondents who “cannot decide” the social acceptance of GEF (Fig 1b) would or would not consume GEF (Fig 1c), and of which, how many were related to trust issues (Fig 1c)? Additionally, how does the degree of trustworthiness (Fig 1c) relate to the information the respondents are interested in (Fig 3)? What is the percentage of well-informed public are acceptive of GEF? Drawing some careful correlation analysis might provide more information and improve discussion.

Answer

We appreciate to this suggestion. We added some analysis and description of results. 

Firstly we added results of correlation of questions shown in Fig1(b) to Fig(1d), and added sentence after the Fig1. This result is shown as Sup2.

There are strong correlations between the answer pattern of the three questions on attitudes toward GEF (See Sup. 2). Generally, the stronger respondents think GEF will be accepted in society, the stronger they agree with eating GEF. Simultaneously, they also show their trust in expert’s discourses on the safety of GEF.

In addition, we also conducted cross-tabulation between trust-worthiness of expert’s on GEF (see Fig. 1(d)), Important factors for acceptance, and interested topics. These results were shown as Sup3 and 4. Description of result were written as below and added after the Fig 4.

We also conducted cross tabulation of public responses between questions on the trust-worthiness of experts on GEF (see Fig. 1(d)), important factors for acceptance, and interested topics (see Sup. 3 and 4). Generally, we found that people who trust expert’s discourses on safety regard scientific validation, necessity, and the prevention of abuse and misuse by regulation as key factors for social acceptance of GEF. They also would like to know the mechanism, benefit, and cost compared to another group. However, people who do not trust expert’s discourses on safety regard risk and clarification of responsibility and liability as key factors for social acceptance of GEF. They would like to know measures for negative effects, national policy or regulation, and ethical issues.

3. Since the Japanese Society of Genome Editing is open to both academic and industry experts, authors should indicate if the experts are from the public or private sector. Reanalysis of the data taking the private or public sector into consideration would provide another perspective. The reason for this is to take into consideration the possibility of bias, i.e., that the private sector could be profit-driven.

Answer

The survey did not ask about the occupational sector, so the exact numbers are not known. However, the majority of genome editing society and the used mailing list were academia.

4. In agreement with the authors’ awareness of the limitations of this survey, perhaps difficult but a follow-up on the public’s acceptance of GEF after the release of GEF products into the Japanese market would be highly interesting. While this request might be out of scope, discussion towards this direction would be appreciated. A helpful literature: https://doi.org/10.3390/su15129662

Answer

We appreciate great suggestion. We added the literature below.

Shigi, R.; Seo, Y. Consumer Acceptance of Genome-Edited Foods in Japan. Sustainability 2023, 15, 9662. https://doi.org/10.3390/su15129662

In addition, we revise our introduction and discussion, considering implication of those literatures. We added sentences below:

A recent literature, after the onset GEF commercialization, showed consumer acceptance of GEF—with 550 respondents [44]. It was found that an interaction existed between awareness of genome-editing technology, information credibility, perceived usefulness, and willingness to buy GEF [44].

And then, we added sentence below to discussion:

However, there is a pertinent need to provide the public with basic information on GEF, considering the potential effects of information sharing on the mechanisms and benefits of GEF suggested by previous studies [44].

Minor comments:

Minor-1: Line 1: Nobel prize

Answer: We revised typos. 

Minor-2: Introduction could be improved, e.g. line 67-71: More elaboration on what are the 3 categories of GEF, and what are the 2 categories of regulation- how are they differently regulated?

Answer

We added explanation about three categories of GEF and their regulatory treatment. We added paragraphs as below in Introduction.

Within the Japanese regulatory system on research and development, genome editing is differently controlled by three types of editing: SDN-1, SDN-2, and SDN-3. SDN-1 is the technique of introducing a break at a target site in the genome. SDN-2 is the addition of a few base pairs of nucleotides to the target site. SDN-3 is introducing a gene at the target site. These types of gene editing are differently regulated according to the viewpoints of environmental assessment and food sanitation [9, 12]. 

For field cultivation of genome-edited crops, they are regulated mainly by the Ministry of Environment (MoE) and Ministry of Agriculture, Forestry, and Fishery (MAFF), and their discussions are related to issues of environmental assessment and biodiversity. In this field cultivation, categories of SND-2 and SDN-3 of genome-edited crops (or animals) are regulated under the Cartagena Domestic Law enacted in conformance with the Cartagena Protocol [9, 12].

Minor-3: Table 1: Under Education “High/Graduate School” should be “High/Graduate school”. Under Annual Income, “4 millionn” should be “4 million”

Answer: We revised typos. 

Minor-4: In general, language support would facilitate a smoother reading process. 

Answer: Our manuscript were native checked by Editage. We attached certification.

Reviewer 2:

1. In abstract, a number of general public is given 3000, whereas this number is 4000 in the methods. Please correct it.

2. Correct Important word in Table 2.

Answer: We revised typos.

---

## [Decision Letter · Decision Letter 1]

14 Feb 2024

PONE-D-23-28736R1A comparative analysis of attitudes toward genome-edited food among Japanese public and scientific communityPLOS ONE

Dear Dr. Shineha,

Thank you for submitting your manuscript to PLOS ONE. After careful consideration, we feel that it has merit but does not fully meet PLOS ONE’s publication criteria as it currently stands. Therefore, we invite you to submit a revised version of the manuscript that addresses the points raised during the review process.

The latest version of the articles address most of the comments by the reviewers. Before finalizing, please address few minor suggestion made by the reviewer. ==============================

We look forward to receiving your revised manuscript.

Kind regards,

Basavantraya N. Devanna, PhD

Academic Editor

PLOS ONE

Journal Requirements:

Additional Editor Comments :

Dear authors,

Thank you very much for the revised version. The improved article is much better and a few suggestions (minor) are made by one of the reviewers.

Regards,

Reviewers' comments:

Reviewer's Responses to Questions

**Comments to the Author**

1. If the authors have adequately addressed your comments raised in a previous round of review and you feel that this manuscript is now acceptable for publication, you may indicate that here to bypass the “Comments to the Author” section, enter your conflict of interest statement in the “Confidential to Editor” section, and submit your "Accept" recommendation.

Reviewer #1: All comments have been addressed

Reviewer #2: All comments have been addressed

2. Is the manuscript technically sound, and do the data support the conclusions?

Reviewer #1: Yes

Reviewer #2: Yes

3. Has the statistical analysis been performed appropriately and rigorously? 

Reviewer #1: Yes

Reviewer #2: Yes

4. Have the authors made all data underlying the findings in their manuscript fully available?

Reviewer #1: Yes

Reviewer #2: Yes

5. Is the manuscript presented in an intelligible fashion and written in standard English?

Reviewer #1: Yes

Reviewer #2: Yes

6. Review Comments to the Author

Reviewer #1: The authors have addressed most of the comments made significant modifications to the manuscript. I especially appreciate Supplemental Table S2 and Table S3 and I invite the authors to consider making Tables S2 and S3 as main tables and/or discuss them more heavily in the Results/Discussion. There are a few minor details (including final version of translated questionnaire) which require polishing before I can recommend for acceptance.

Minor

Line 107: SDN-2, not SND-2

Line 146: Please change “images” to “impression”

Line 324: Please check if ‘sup. 3’ is supposed to be Table S2. Provide table legends for Table S2 and Table S3. Please change “Generally, we found that people who trust..” to “Generally, we found that people trust..”

Line 327: Please revise this line to make the message clearer.

Line 328: Please change “people who do not trust..” to “people do not trust”

Line 330-332: Please revise this line to make the message clearer.

Line 322-331: Please use quotation marks for each factor, i.e. “scientific validation”, “necessity” etc, to ease reading. Consider revising this paragraph- message not easily understood.

All figures : Make sure the legends in the main text are similar to those used in the figures.

Methods: Please describe how statistics and correlation analysis were performed.

Reviewer #2: The changes asked to make are incorporated in the manuscript entitled, 'A comparative analysis of attitudes towards genome-edited food among Japanese public and scientific community'.

7. PLOS authors have the option to publish the peer review history of their article (what does this mean?). If published, this will include your full peer review and any attached files.

Reviewer #1: No

Reviewer #2: **Yes: **Amolkumar U. Solanke

---

## [Author Response · Author response to Decision Letter 1]

15 Feb 2024

Response to reviewer’s comment

We would like to appreciate to reviewers for their valuable opinions to our manuscript. We would like to show summary of our revision for reviewer’s comment here. In following text, reviewer comments were shown with Bold.

Reviewer 1:

The authors have addressed most of the comments made significant modifications to the manuscript. I especially appreciate Supplemental Table S2 and Table S3 and I invite the authors to consider making Tables S2 and S3 as main tables and/or discuss them more heavily in the Results/Discussion. There are a few minor details which require polishing before I can recommend for acceptance. 

Answer: We revised our manuscript according to this reviewer’s suggestion. We moved Table S2 and S3 to main Tables (as Table 3 & 4), and added below sentences in the “Discussion” section. 

Simultaneously, we need to consider the different attitudes between the two types of groups, with or without trust in experts’ discourses on safety. Tables 3 and 4 show that the two groups have different interests and concerns regarding GEF. People who trust expert’s discourses on safety generally have interests in scientific contents and “the way of prevention of abuse and misuse by regulation.” On the other hand, another group have interests in issues related risk governance such as “whether experts can deal with risks,” “clarification of responsibility and liability,” “national policy or regulation,” “ethical issues,” and so on. We should not see the public as a monolith. We need to consider what kinds of information and factors are of interest and expected in each group.

Minor

Line 107: SDN-2, not SND-2

Answer: We revised typos. 

Line 146: Please change “images” to “impression”

Answer: We revised it. 

Line 324: Please check if ‘sup. 3’ is supposed to be Table S2. Provide table legends for Table S2 and Table S3. Please change “Generally, we found that people who trust..” to “Generally, we found that people trust..”

Answer: We revised it. And we added table legends to Table S2 and Table S3. In addition, we move supplemental tables to main tables according to reviewer’s suggestion. Revised manuscripts are also shown above.

Line 327: Please revise this line to make the message clearer.

Answer: We revised this line as below:

For more effective communication between the public and experts on GEF, 

Line 328: Please change “people who do not trust..” to “people do not trust”

Answer: We think relative pronoun “who” is necessary here. We revised paragraphs around this line according to previous reviewer’s suggestion regarding Table S2 and S3. Revised sentences are shown in before and file with markup.

Line 330-332: Please revise this line to make the message clearer.

Answer: We revised this line as below:

It is important to provide basic information on GEF to the public continuously, considering the potential effects of information-sharing on the mechanisms and benefits of GEF, as suggested by previous studies.

Line 322-331: Please use quotation marks for each factor, i.e. “scientific validation”, “necessity” etc, to ease reading. Consider revising this paragraph- message not easily understood.

Answer: We revised it. 

All figures : Make sure the legends in the main text are similar to those used in the figures.

Answer: We re-checked and revised them.

Methods: Please describe how statistics and correlation analysis were performed. 

Answer: We added sentence below in the “Methods” section.

In our statistical analysis, we conducted a chi-square test to determine independence. In addition, we calculated Pearson’s correlation (see Table 3). For our analysis, we used SPSS ver. 26.

---

## [Editor Report · Decision Letter 2]

23 Feb 2024

A comparative analysis of attitudes toward genome-edited food among Japanese public and scientific community

PONE-D-23-28736R2

Dear Dr. Shineha,

We’re pleased to inform you that your manuscript has been judged scientifically suitable for publication and will be formally accepted for publication once it meets all outstanding technical requirements.

Kind regards,

Basavantraya N. Devanna, PhD

Academic Editor

PLOS ONE

Additional Editor Comments (optional):

Thank you for including and working on the suggestions. The current version of the article is more polished.
---

## [Editor Report · Acceptance letter]

27 Feb 2024

PONE-D-23-28736R2 

PLOS ONE

Dear Dr. Shineha, 

I'm pleased to inform you that your manuscript has been deemed suitable for publication in PLOS ONE. Congratulations! Your manuscript is now being handed over to our production team.

Kind regards, 

on behalf of

Dr. Basavantraya N. Devanna 

Academic Editor

PLOS ONE